# 140 GHz Ultra-Long Bessel–Like Beam with Near-Wavelength Beamwidth

**DOI:** 10.3390/s20236791

**Published:** 2020-11-27

**Authors:** Gyeongsik Ok, Kee Jai Park

**Affiliations:** Department of Research Group of Consumer Safety, Korea Food Research Institute, Wanjugun 55365, Korea; jake@kfri.re.kr

**Keywords:** Bessel–like beam, axicon lens, depth of focus, near-wavelength resolution, sub-terahertz frequency, nondestructive inspection

## Abstract

The Bessel–Gauss beam has outstanding features, such as long depth of focus (DOF) and super resolution for nondestructive imaging inspection. However, most approaches for generating a nondiffractive beam have mainly focused on extending the DOF. In this study, the ultra-long high-resolution Bessel–like beam was first demonstrated in a sub-THz wave range (140 GHz). An axicon lens having an apex angle of 110° was used to generate the highly focused Bessel–like beam. To extend the depth of focus, we varied the incident beam angle on the axicon by moving the first lens distance. With the newly developed beam profiler, 3D beam profiles were acquired for characterizing in detail the beam propagation. As a result, even if the depth of focus was 72 times (154 mm) the source wavelength (2.143 mm), the focusing beamwidth was simultaneously maintained at 1.4 times (3.0 mm) the wavelength (i.e., the near-wavelength beamwidth). An ultra-long needle beam of near-wavelength size can promote the applicability of the sub-THz imaging technique in noninvasive sensing applications, such as computer tomography, materials inspection, and through-the-wall-imaging.

## 1. Introduction

Sub-THz waves (mm waves, 30–300 GHz) and THz waves (0.1–10 THz) cannot penetrate metalized film packaging or high-water-content products. Rather, they are transparent to most dry dielectric materials [1,2]. Hence, sub-THz and THz waves offer many benefits to nondestructive inspection applications. First, sub-THz and THz waves are nonionizing radiation, which has much lower energy than X-rays and is safe for biomaterials, such as foods and humans. Second, most molecules have specific spectroscopic fingerprint patterns in those bands, extending between the microwave and infrared spectra [3,4]. In particular, considering that power attenuation in sub-THz ranges caused by water content is slightly lower than that in the upper bands, continuous-wave (CW) imaging in these bands can be advantageous for various materials inspection. This is because a better signal-to-noise-ratio (SNR) exists in the lower bands [5]. Unfortunately, the wavelength of the sub-THz range is longer than that of the upper THz band. As a result, the image resolution can be degraded in these frequency ranges. Despite the limitations, monochromatic CW sub-THz imaging is the most practical and useful for satisfying industrial requirements, such as low-cost high-speed detection and high transmission power [5,6,7,8]. 

The CW Bessel–Gauss beam, a non-diffractive beam, has been utilized to overcome the spatial resolution hurdles of low-frequency sub-THz sources [9]. Based on Bessel–Gauss beam focusing and mechanical raster scanning, sub-THz transmission imaging can provide sub-wavelength resolution and extended depth of focus (DOF). Because the most prominent feature of the Bessel–Gauss beam is its prolonged DOF, its needle beam [10], which has a DOF measuring tens-of-wavelengths [9], can still be unsatisfactory for a few nondestructive inspection applications. For instance, THz computer tomography (CT) imaging requires more extended DOF as well as high spatial resolution to accurately examine the internal structures of given samples [8,11,12,13]. However, owing to the Gaussian beam property (e.g., Rayleigh range), a tradeoff exists between the DOF and beam-spot size in conventional THz beam-focusing applications. In contrast, THz CT imaging by the Bessel–Gauss beam can provide outstanding imaging performance by retaining super-resolution over a longer DOF. Despite the superior dual characteristics of the Bessel–Gauss beam, many previous studies on THz CT imaging have focused mainly on extended DOF characteristics in this band [8,11,12,13]. 

Various approaches have been developed to extend the DOF of the non-diffractive beam (e.g., axicon lens, narrow annular aperture filter, ring-lens, and computer-generated hologram) [11,12,13,14,15,16,17,18]. Among these methods, refractive axicons have better energy efficiency, considering the SNR [19,20,21]. As a result, axicons have been widely employed for generating the Bessel–Gauss beam in sub-THz and THz frequency ranges [7,9,11,12,13]. The DOF range behind the axicon is inversely proportional to the cone angles of the propagating waves and proportional to the input beam radius. Thus, to extend the DOF, the incident beam size should be larger or the cone angle shallower. Considering practical optic sizes, extending the DOF by the cone angles is more preferable. The concept of the cone angles decreasing during beam propagation has been previously reported with respect to Bessel–like beams having *z*-dependent cone angles, which are generated by a combination of conventional optical elements [19,20,21]. However, studies on the ultra-long DOF having a near-wavelength beam width have not been reported in the sub-THz and THz band. Thus, based on the concept of Bessel–like beams with varied cone angles, we attempt to generate ultra-long beams with near-wavelength beamwidths in the sub-THz band. We utilize two lens pieces (i.e., aspherical and axicon) to obtain *z*-dependent cone angles. The range of the angles is tuned with the incident beam angle on the axicon by moving the first lens distance from the source. To maintain the near-wavelength beamwidth, a low-apex angle axicon lens is used.

The remainder of this paper is organized as follows. In Section 2, we describe the ray-tracing simulations of axicon-generated Bessel–like beams to demonstrate the means by which its DOF can be controlled. In Section 3, we present Bessel–like beams having various DOFs that are generated from a polyethylene (PE) axicon at 140 GHz. Their beam profiles are experimentally measured with a custom-made beam profiler to confirm the near-wavelength resolution and considerably prolonged DOF. In Section 4, the results and discussion are provided, and conclusions are given in Section 5.

## 2. Ray-Tracing Simulations of Axicon-Generated Bessel–Like Beams

The DOF ($Z_{max}$) of the axicon-generated Bessel–Gauss beam can typically be varied with two geometrical parameters, such as the axicon apex angle and the waist of the incident Gaussian beam, as described in the following equations and Figure 1a:(1)$$Z_{max}=\frac{\omega_{0}}{tan\;\alpha_{0}},$$
(2)$$\alpha_{0}=arcsin\left(\frac{n}{n_{0}}cos\left(\frac{\tau }{2}\right)\right)+\frac{\tau -\pi }{2},\left(0<\alpha_{0}<\frac{\tau }{2}\right).$$
Here, $\omega_{0}$ is the beam radius of the incident beam, and $\alpha_{0}$ is the semi-apex angle of the focused beam, which is related to the following axicon parameters: the refractive index of the axicon (PE, 1.523), *n*; the refractive index of the surrounding medium (air, 1.0), $n_{0}$; and the apex angle of the axicon, *τ*. Moreover, the full-width at half-maximum (FWHM) size ($\rho_{FWHM}$) of the Bessel–Gauss beam can be expressed in Equation (3) as an image-resolution indicator [22]:(3)$$\rho_{FWHM}=2\frac{1.1264}{\frac{2\pi }{\lambda }\sin\alpha_{0}},\;$$
where λ is the source wavelength (=2.143 mm). The sub-THz source (divergence angle, *θ*, ~30°) in Figure 1a is a 140-GHz IMPact ionization Avalanche Transit-time (IMPATT) diode (Terasense Group Inc., San Jose, CA, USA). We set the apex angle of the axicon, *τ*, to 110° for near-wavelength beam focusing. However, all lenses had radii (R) of 50 mm.

Usually, with a thin axicon having a large apex angle, Equation (1) is used to estimate the DOF. However, when the axicon apex angle becomes acute, a corrected equation should be used for accuracy:(4)$$Z{’}_{max}=\frac{\omega_{1}}{tan\;\alpha_{0}}-\frac{\omega_{1}}{tan\;\frac{\tau }{2}}\left(=Z_{offset}\right).\;$$
Here, $Z_{offset}$ is the overestimated length inside the axicon from the overall beam length ($Z_{offset}$). $\omega_{0}$ is the beam radius on the incident surface of the axicon, whereas $\omega_{1}$ is on the exit surface. When the incident beam is collimated, $\omega_{0}\;$and $\omega_{1}$ are the same, as shown in Figure 1a. In contrast, when the incident angle is not zero on the incident surface, the beam radius becomes different and varied with L variations. Moreover, $\omega_{1}\;$is a critical parameter for calculating the precise DOF, as shown in Equation (4) and Figure 1b.

In the case where the incident beam on the axicon surface is collimated, using the above Equations (3) and (4) and the geometrical parameters, the DOF and beamwidth of the Bessel–Gauss beam can be easily estimated. For instance, $Z_{offset}\;$can be approximated to 50 mm using Equation (1), whereas $Z{’}_{max}$ is precisely calculated to 33 mm. It implies that $Z_{offset}\;$ is 17 mm, and $Z{’}_{max}$ is a more accurate value of the DOF. Angle $\alpha_{0}$ is calculated as 25.9° by Equation (2). In turn, the theoretical focusing beamwidth is calculated as 1.76 mm by Equation (3).

Next, by further increasing the DOF from the collimated case, we employed a diverging beam on the axicon surface. Its incident angle on the axicon was manipulated by varying the distance (L) between the source flange and the first lens (designed back focal length = 72.5 mm, PE), where the axicon lens was spaced 60-mm away from the first lens surface. As a result of reducing the distance, L, from the collimated beam, Bessel–like beams having *z*-dependent cone angles can be generated, so that the DOF can be extended. Unfortunately, Equation (2), which obtains the cone angle, $\alpha_{0}$, cannot be applied in this case. Hence ray-tracing simulations should be implemented for calculating the cone angle, $\alpha_{0}$, with varying distance L from the collimated beam, where the cone angle is that of the marginal ray. Using the calculated cone angles from the ray-tracing, the beam diameter, $\rho_{FWHM}$, and the exact DOF, $Z{’}_{max}$, can be obtained from Equations (3) and (4). To calculate the $Z{’}_{max}$, the beam radius, $\omega_{1}$, to varied L should also be obtained from the ray-tracing. Additionally, to investigate the effect of the beam radius, $\omega_{1}$, changes on the DOF variation, $Z{”}_{max},\;$and the incident angle are shown in Figure 1a, also calculated from the simulations. In particular, $Z{”}_{max}$ is obtained from Equation (4) based on the assumption that the incident beam radius, $\omega_{1}$, is constant for varying L. By comparing $Z{’}_{max}$ and $Z{”}_{max}$, the major factor to affect the extended DOF can then be examined.

When the incident beam is collimated, because the rays both adjacent to and away from the apex are parallel, the uniform focusing beam width of Equation (3) is expected during beam propagation. In contrast, when increasing the incident angle on the axicon, the farther the rays are from the apex of the axicon, the corresponding cone angles become more *z*-dependent, as shown in Figure 1b. The cone angle, $\alpha_{0}$, decreases with the distance from the apex of the axicon, and Equation (2) cannot be applied to the case, as discussed above. Moreover, Equation (3) can be valid when the wavefront of the incident beam on the axicon is a plane wave [22]. On the contrary, this may be difficult to apply when the wavefront at the incident surface is spherical or parabolic. However, for roughly investigating the beamwidth change by the cone angles, it can be estimated with Equation (3) by assuming that the wavefront on the incident surface is a plane wave.

For obtaining the full range of the *z*-dependent cone angles, the half-source divergence angle, *θ*, was varied from 15° (marginal rays) to 1° (paraxial rays) at 2° increments for various lens distances, L. As a result, the range of marginal rays corresponding to each half-source angle can provide each cone angle. Thus, the FWHM variations on the propagation axis within the DOF range can be estimated for a given distance, L.

## 3. Measurement Setup for Focusing Beam Profiles 

To visualize the sub-THz focusing beam, a three-axis beam-profile measurement setup, which is faster than a previous system [7,9], was developed, as shown in Figure 2. In the visible and infrared ranges, the beam profiles could be easily measured using a commercially available beam-profiler (e.g., a charged coupled-device camera). However, because there was no method to measure the cross-sectional profile for a long-range (several tens of cm), a custom-made measurement setup was developed as follows.

The X-axis line-scan shown in Figure 2 was performed using a high-speed linear motor stage (Jenny Science AG, Rain, Switzerland). Y- and *Z*-axis scan stage were constructed with a ball-screw-type motor (DST Robot, Chungcheongnam-do, Korea). A Schottky diode detector (Virginia Diodes Inc., Charlottesville, VA, USA) having a 3 mm pinhole was mounted on the vertical scan stage for raster scanning. An optical rail with various optic components was positioned along the *Z*-axis on an optical table, as depicted in Figure 2. 

A 2D image corresponding to a 2D beam intensity profile was acquired by a fast axis line scan movement incorporated with a slow axis line scan, where the scan resolutions of both line scans were 0.2 mm. Each pixel intensity value on the screen was fed from a National Instruments (NI) data acquisition (DAQ) board (NI 6366, NI, Austin, TX, USA), which was synchronized using stage encoder signals. Before transmission to the DAQ board, the analog signals from the detector were amplified by a low-noise preamplifier (SR 560; Stanford Research Systems, Sunnyvale, CA, USA). By using the fast 3D beam profiler (Table 1), three types of surface beam profiles could be acquired: X–Y scan surface, X–Z scan surface, and Z–Y scan surface.

## 4. Results and Discussion

### 4.1. Ray-Tracing Simulation Results on the Bessel–Like Beam Shaping

Using the ray-tracing calculation of Section 2, both incident-beam radii on the incident and exit surfaces were calculated. Additionally, the incident angle on the axicon and the cone angle, $\alpha_{0}$, corresponding to the beam radius on the exit surface, was calculated for several distances, L, as shown in Figure 3a,b. In particular, because the source was not a point, the distance, L, between the source flange and the first lens surface was defined as 67 mm in the case of the collimated beam, except that it was 5.5 mm from the designed back focal length of 72.5 mm. L ranged from 67 to 21 mm in 5-mm increments. Within this range, some L were identically adjusted to the measured distance. The calculation results are shown in Figure 3.

As expected, both the beam radius and the incident angle varied simultaneously on the axicon surface when the first lens distance, L, varied within the ranges. Figure 3a shows that, even if the lens distance, L, was decreased by one third from the collimated L = 67 to L = 21 mm beam, its radius, $\omega_{1}$, on the exit surface would be reduced by only 2 mm (8%). Instead, the incident and cone angles of the marginal ray were rapidly changed, as shown in Figure 3b. Using the beam radius and cone angle, the DOF values from Equation (4) were calculated in Figure 3c. As the distance, L, reduced from 67 to 21 mm, the cone angle decreased by 10.8°. Hence, the DOF of the Bessel–like beam doubled from 33 to 67 mm. Furthermore, when varying the L, the $Z{”}_{max}$ in Figure 3c was calculated based on the assumption that the beam radius, $\omega_{1}$, on the exit surface was constant, whereas $Z{’}_{max}\;$was calculated based upon a varied beam radius,$\;\omega_{1}$. Figure 3c shows that two curves were almost identical, implying that the incident-angle change could be a major factor affecting the extended DOF as it corresponds to the reduced marginal cone angle. 

Using Equation (3) and the parameters of Figure 3a,b, the focusing beamwidth was also estimated, as shown in Figure 3d. When L = 67 mm (collimated), the beamwidth was easily estimated using Equation (3). However, when decreasing the lens distance, L, the cone angles within the range had a range of values (not singular), as discussed. To calculate the full ranges of the cone angle for a given distance, L, the half-source angles for each marginal ray were varied from 15 to 1° in 2° increments. Each marginal ray corresponding to the given half-source angle constructed the cone angles via ray-tracing. The FWHM was then obtained using Equation (3). Among the results, the variations of FHWM for four internal angles is shown in Figure 3d. At the collimated distance, L, all cone angles were parallel. Thus, all FWHM values were identical. In contrast, with a decreasing L, the non-parallel inner rays formed different cone angles, and the FWHM of each is shown in Figure 3d. This shows that the FWHM difference between internal angles increased with a decreasing L. Thus, the farther away from the apex the axicon, the wider the beamwidth. To compare the experimental results, simulation results are discussed next. Although not shown in the graphs, in the extreme case where L was 0, the calculation result shows that the DOF was 94.8 mm (almost tripled), and the beam width as 4.08 mm (only 1.9 wavelengths). Therefore, the beamwidth theoretically does not exceed two wavelengths in tandem with more than twice the DOF.

### 4.2. Beam-Profile Measurement Results on the Bessel–Like Beam Propagation

To visualize the actual Bessel–Gauss beam propagation, we measured the beam profiles behind the axicon by using the 3D beam profiler depicted in Figure 2. For the ray-tracing results of Figure 3, we adjusted the first lens distance, L, from the 140-GHz source flange to construct several needle beams (L = 67, 56, 51, 46, 41, and 36 mm). 

As an example of the measured beam profiles, Figure 4 shows a 3D Bessel–like beam propagation image for L = 46 mm among the various needle beams. The 3D beam propagation image consists of three 2D beam profile surfaces, including the YZ, XZ, and XY scan surfaces. Here, the XY surface was chosen at Z = 82 mm with a maximum intensity along the *Z*-axis. The YZ and XZ surfaces were examined at a 0.2-mm resolution over 40 × 200 mm (~19 min for 200 × 1000 pixels). The XY surface was measured with a 0.2-mm resolution over 40 × 40 mm (~3 min for 200 × 200 pixels). 

As expected in the ray-tracing simulation, the beamwidth of the Bessel–like beam broadened with the beam propagation, and the side-lobe peak positions varied. However, the measured DOF was more extended than in the simulation. Moreover, there were oscillatory interference patterns in the propagation profiles. This was caused by the standing-wave effect generated from reflections on the pinhole metal surface in front of the detector. 

Next, to obtain precise extended DOF values, we extracted the axial intensity profile at Y = 0 along the *Z*-axis from the YZ scan surface. As a reference beam, the collimated Bessel–Gauss beam profiles at L = 67 mm are described in Figure 5a. The axial intensity profile of the main lobe is shown in Figure 5b. Owing to the oscillating patterns of the standing waves (inset graph in Figure 5), the numerical approximation for precise DOF estimation was obtained using a locally weighted scatterplot smoothing regression, shown in Figure 5b. For examining the oscillatory patterns in depth, beam profiles were measured repeatedly five times. It shows that, even with repeated measurements, the amplitude and phase of the oscillating patterns hardly changed. Moreover, it shows that the intensity curves fluctuated with a period of about half the wavelength, as seen in the inset graph. However, at around Z = 15 mm, the envelope of their amplitudes decreased rapidly and increased again. This was caused by the source intensity distribution not being a perfect Gaussian. To investigate the irregularity in the propagation profile, the 2D spatial intensity profile of the IMPATT source was also measured 12-mm away from the flange of the source, as shown in Figure 6.

Figure 6 shows that the intensity distribution of the source differed noticeably from that of a perfect Gaussian beam, and the intensity was distributed asymmetrically along the horizontal direction. Furthermore, when comparing the used wavelength with the dimension of the optical system, the diffraction at the edge of the lens clearly affected the beam propagation after passing through the lenses. They strongly affected the beam intensity distribution at every surface through the lenses. Therefore, the axial intensity distribution differed from that of the ideal Bessel–Gauss beam.

Based on the prior smoothing regression method, all axial intensity profiles were compared, as shown in Figure 7. All curves were obtained from the measured spatial 2D beam profiles on the YZ surface and smoothed using the regression method. As L decreases, six different axial profiles are displayed in the figure. Owing to the irregular intensity distribution of the source, multiple peaks appeared along the beam propagation. Their estimated DOF values are summarized in Table 2.

Next, the FWHM values of the Bessel–like beam were investigated by comparing the ray-tracing results with the experimental ones. As discussed, the FWHM variations along the beam propagation direction were affected by the range of the cone angle at a certain distance, L, where the full ranges of the cone angle were obtained from the ray-tracing simulations for a given lens distance (L = 67, 56, 51, 46, 41, and 36 mm). Thereby, the FWHM values were calculated using the cone angles of Equation (3). The Z coordinates corresponding to each cone angles were calculated using Equation (4). The FWHM results are displayed in Figure 8, where it can be seen that the calculated beamwidth values had sub-wavelength sizes in most cases. In Table 2, the averaged beamwidth values over the entire Z range are summarized for several L. In particular, for the diverging beam, the slope of the curves monotonically increased as the distance, L, decreased. This shows that the beamwidth difference near to and far from the axicon got wider as L decreased.

Similar to Figure 8, Figure 9 shows various measured FWHM values within the DOF ranges. The raw FWHM curves had slight oscillatory patterns, like the axial intensity distribution of Figure 5b. Thus, the curves were also smoothed as in Figure 9. Similar to the ray-tracing results, the slope of the curves increased with the decreasing lens distance, L. In most measurement results, the curves showed that the FWHM values considerably fluctuated near the apex of the axicon, similar to the axial intensity distribution profiles. Thus, the fluctuation may have been be caused by irregular intensity distributions in the central area of the light source, considering the ray-path from the source. Compared with the ray-tracing results, the beamwidth size in Figure 9 increased significantly, but it did not exceed twice the wavelength. Similarly, the average values of FWHM over the entire Z range is displayed in Table 2.

As summarized in Table 2, the distance, L, was reduced, whereas the measured DOF and beamwidth were considerably extended. Despite the extended beamwidth, it remained within 1.4 wavelengths, whereas the DOF increased 7.3 times when the L of the first lens was varied from 67 to 36 mm. As described, this resulted in a much-longer measured DOF, compared with the geometrical ray-tracing estimates. This discrepancy can be explained by the following backgrounds. When comparing the used wavelength with the dimension of the beam waist and lenses, the diffraction effect had great importance on the beam propagation in the sub-THz band. From the diffraction effect, geometrical ray-tracing could not fully describe the beam shape and propagation in this band. Considering the significant diffraction, the quasi-optical analysis and geometric ray-tracing must be employed to precisely estimate the beam shape [23]. Moreover, from the ray-tracing and the equations, it was assumed that the rays came from a point source, which is not the case in reality. As shown, the intensity distribution of the source was asymmetric and irregular. Because the source was not a perfect Gaussian, the propagating waves from the source had nonuniform divergence angles and an irregular wavefront. Thus, the cone angles of the marginal rays in the measured profiles differed from the expected values. Therefore, the discrepancy in the DOF and FWHM estimation can be explained. 

To maintain the near-wavelength resolution while enabling a very long DOF, it was essential to first select the lens aperture size and axicon apex angle. Then, the prolonged DOF was obtained by increasing the angle of the incident beam. As a result, even if the DOF was significantly extended, the beam width could be maintained near the wavelength size.

The realization of the needle beam had great importance in sub-THz CW imaging. As described, the power attenuation in the sub-THz band was more favorable for transmission imaging than the upper band, even when the image resolution was decreased in the lower band. Moreover, the high-power CW source of the lower band was less expensive than that of the higher band. With these lower-band advantages, the ultra-long and sharp needle beam overcame the imaging resolution limitation and provided deep-structure imaging on high-water content materials. Hence, it provides opportunities for various nondestructive inspection applications, such as computer tomography, materials inspection, and through-the-wall-imaging [24]. 

As exhibited in the 3D beam intensity profiles, the CW sub-THz focusing beam was highly coherent. Hence, a precise lens alignment was required for high-quality imaging. Notably, the beam propagation in sub-THz and THz bands cannot be observed by the naked eye. The invisible beam propagation, therefore, makes optics alignment difficult and time-consuming. To address the experimental limitations of CW imaging, a custom-made 3D beam profiler was developed. Although the interference patterns were not removed in this study, it may be resolved by using mm-wave absorbing materials on the source and detector in this band. Owing to the fast visualization of the beam propagation, the lens design and alignment were considerably easier. In the future, the transmitted beam should also be examined by the beam profiler to better understanding the forwardly scattered beams behind the test object.

## 5. Conclusions

When nondestructively inspecting an object, a long working distance can be a key performance parameter. Thus, an extended DOF is critical to high-quality inspection (e.g., CT). Additionally, the beamwidth focus determines the imaging resolution of a transmission image. It is, therefore, critical to implement a needle beam with an extended DOF and a near-wavelength resolution for those applications. In this study, by controlling the defocusing distance from the collimated case, the DOF of the Bessel–like beam was significantly extended to 72× (154 mm) the wavelength (2.143 mm). The diverging beam was produced by distancing the first lens distance from the source flange (36 mm), where the first lens distance was moved from 67 to 21 mm. Meanwhile, its focusing beamwidth was maintained at almost 1.4× (3.0 mm) the wavelength of the near-wavelength resolution.

Many previous studies in THz imaging have reportedly extended the DOF or separately improved the resolution. However, the present experimental demonstration, presented here for the first time, simultaneously satisfied both properties. Furthermore, apart from the coherent nature of the CW 140-GHz source, the beam propagation was visualized and characterized using a custom-made 3D beam profiler.

A narrow Bessel–like beam with an ultra-long DOF can expand the applicability of the sub-THz imaging to support the inspection of various previously opaque materials. In particular, the combination of those beam properties can provide great opportunities for 3D CT imaging applications.

## Figures and Tables

**Figure 1 sensors-20-06791-f001:**
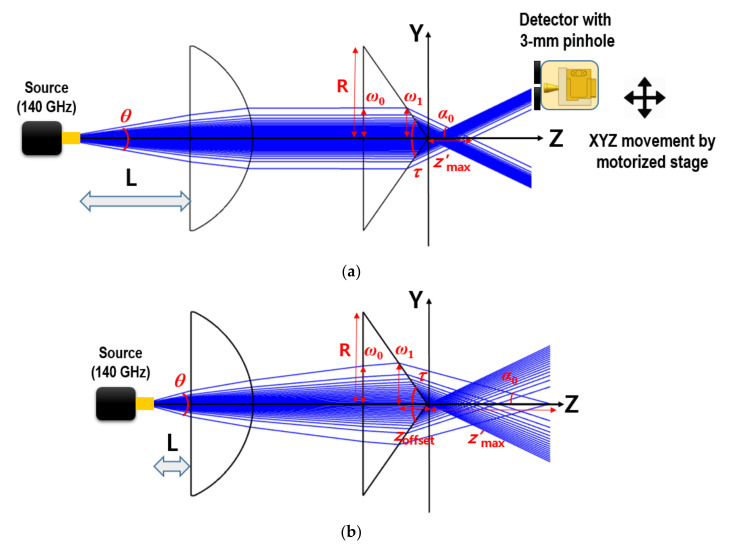
Ray optics simulations (Opticstudio, Zemax, Kirkland, WA, USA) for estimating the DOF of the Bessel–like beam: Bessel–Gauss beam generation (**a**) by a collimated beam on the axicon; (**b**) Bessel–like beam by a diverging beam on the axicon.

**Figure 2 sensors-20-06791-f002:**
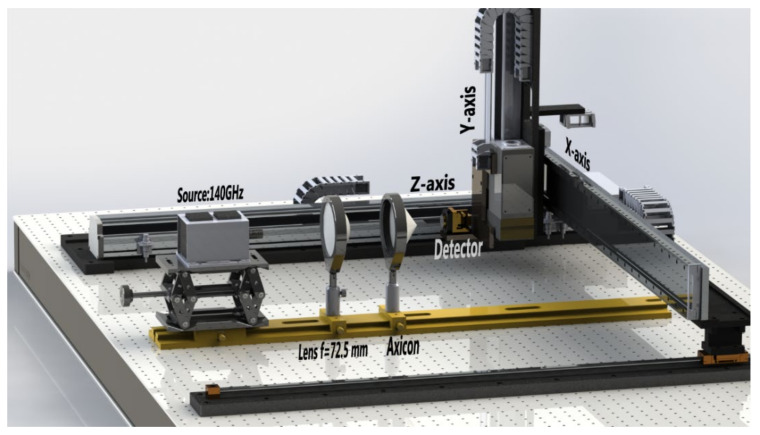
Experimental setup for measuring the Bessel–Gauss beam profiles of the axicon lens.

**Figure 3 sensors-20-06791-f003:**
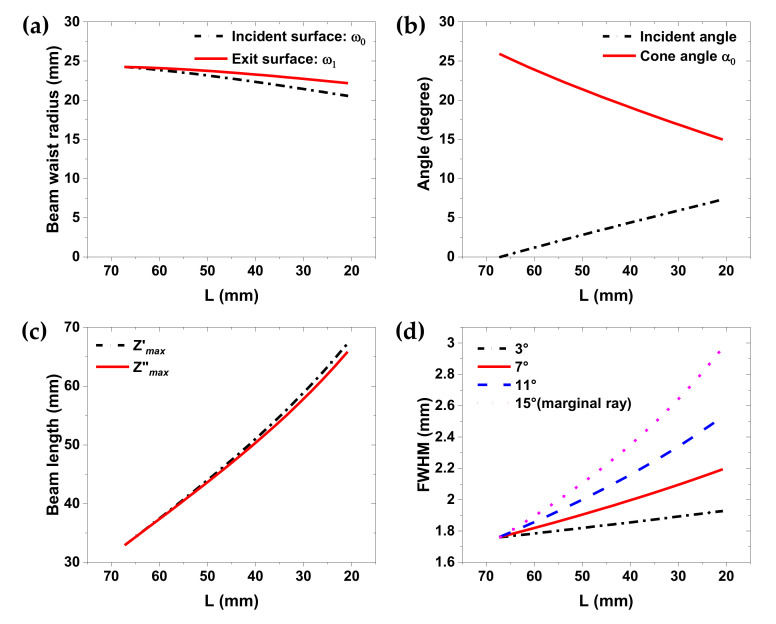
Ray-tracing results for obtaining beam parameters: (**a**) beam-waist radius on the incident and exit surfaces of the axicon; (**b**) incident angle on the axicon surface and cone angle of the marginal ray; (**c**) DOF $Z{’}_{max}\;$by varied$\;\omega_{1}$, and $Z{”}_{max}$ with a constant $\omega_{1}$; (**d**) FWHM values for several internal cone angles, calculated with corresponding half-source angles (=*θ*/2), for varying distances, L, respectively.

**Figure 4 sensors-20-06791-f004:**
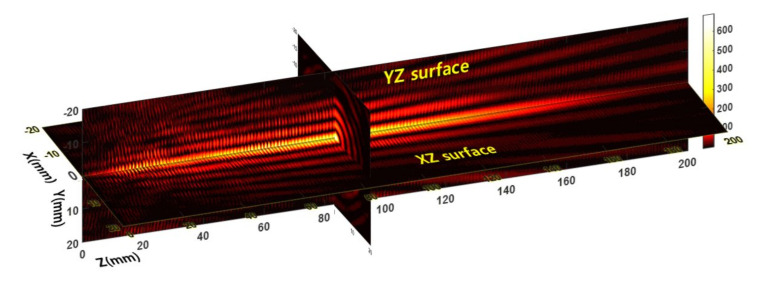
3D Bessel–like beam propagation image obtained from the beam profiler in the case of the 46-mm distance between the source flange and the first lens surface (legend values are arbitrary).

**Figure 5 sensors-20-06791-f005:**
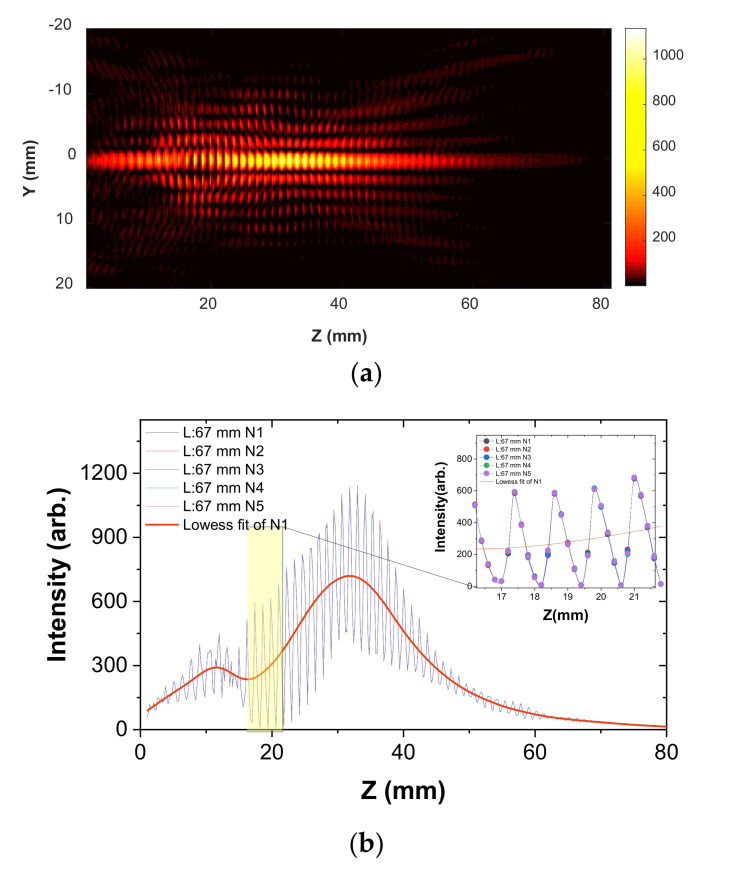
(**a**) Spatial 2D intensity profile on YZ surface measured at the L = 67-mm distance; (**b**) Axial intensity profile along the *Z*-axis and magnified profile (inset) extracted from the 2D profile.

**Figure 6 sensors-20-06791-f006:**
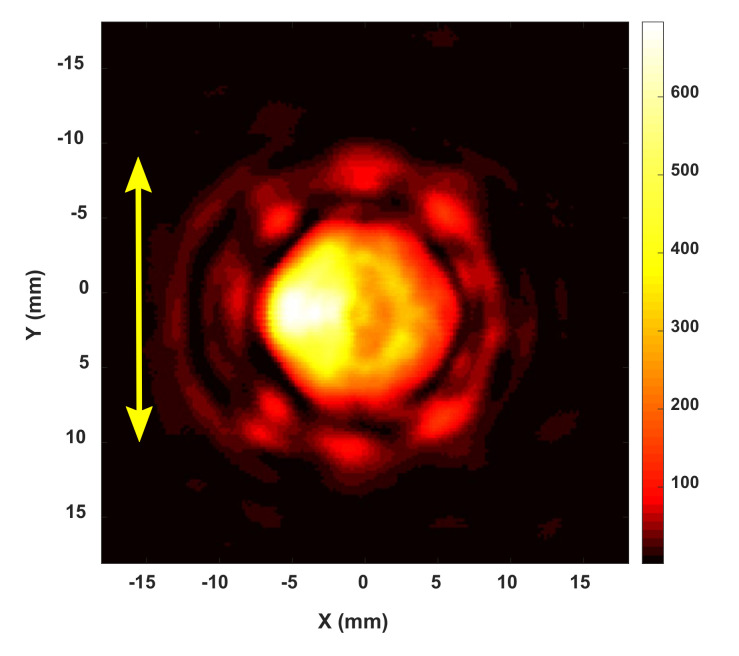
Spatial 2D intensity profile on the XY surface measured 12-mm away from the flange of the IMPATT sub-THz source (yellow line indicates the polarization direction).

**Figure 7 sensors-20-06791-f007:**
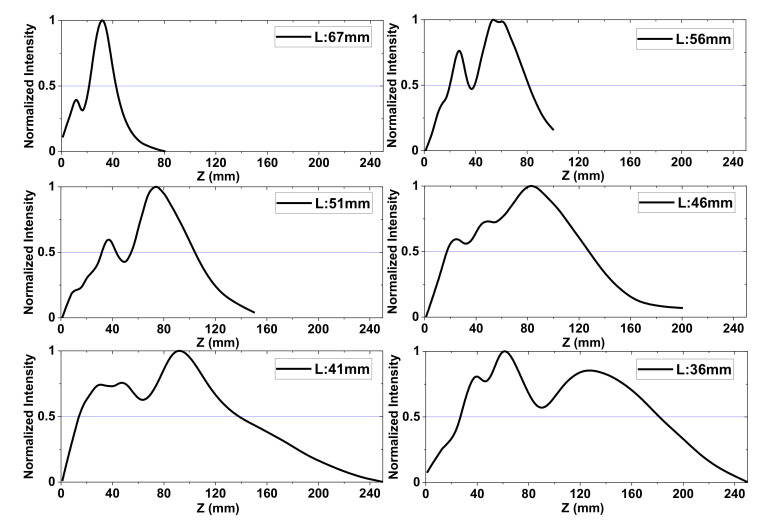
Smoothed axial intensity profiles for estimating the DOF (L = 67, 56, 51, 46, 41, and 36 mm). All curves were normalized to maximum intensity. Each DOF values of the figures are summarized in Table 2.

**Figure 8 sensors-20-06791-f008:**
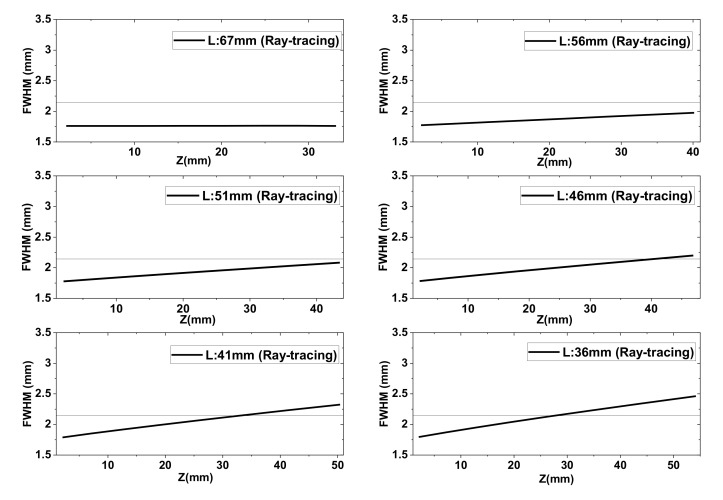
FWHM variations along the *Z*-axis obtained from ray-tracing and Equation (3) (L = 67, 56, 51, 46, 41, and 36 mm). Reference line denotes the source wavelength (2.143 mm).

**Figure 9 sensors-20-06791-f009:**
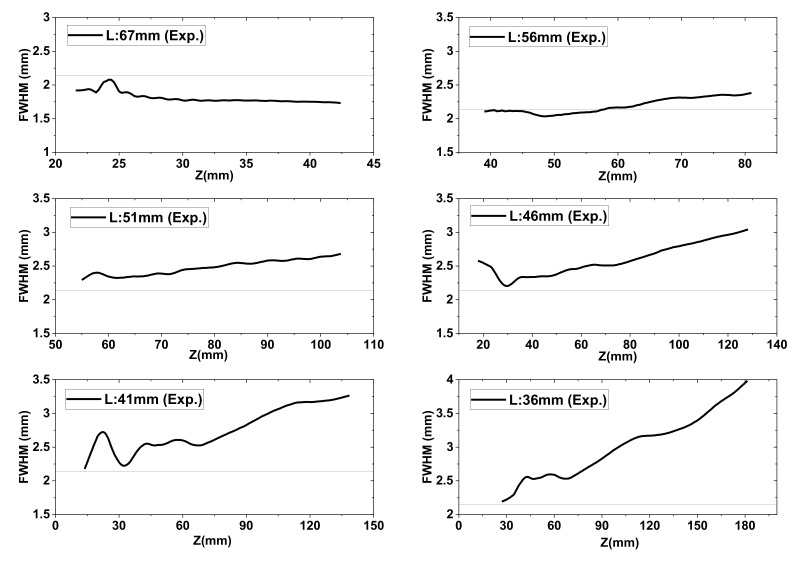
FWHM variations along the *Z*-axis directly obtained from measured 2D intensity profiles (L = 67, 56, 51, 46, 41, and 36 mm). Reference line denotes the source wavelength (2.143 mm).

**Table 1 sensors-20-06791-t001:** Technical specifications of the 3D beam profiler.

Axis	Maximum Scan Length	Minimum Scan Resolution	Maximum Scan Speed
X-axis	800 mm	1.0 μm	2 m/s
Y-axis	300 mm	1.125 μm	1 m/s
Z-axis	800 mm	2.5 μm	1 m/s

**Table 2 sensors-20-06791-t002:** DOF and beamwidth obtained from various beam propagation profiles.

Lens Distance (L)	DOF		Beamwidth (Mean Values)			
	Ray Tracing	Measurement	Ray Tracing		Measurement	
67 mm	33 mm	21 mm	1.76 mm	0.82 λ	1.81 mm	0.84 λ
56 mm	40 mm	61 mm	1.87 mm	0.87 λ	2.19 mm	1.02 λ
51 mm	43 mm	70 mm	1.92 mm	0.90 λ	2.48 mm	1.16 λ
46 mm	47 mm	109 mm	1.98 mm	0.92 λ	2.60 mm	1.21 λ
41 mm	50 mm	123 mm	2.04 mm	0.95 λ	2.77 mm	1.29 λ
36 mm	54 mm	154 mm	2.10 mm	0.98 λ	3.01 mm	1.40 λ

(λ: Source wavelength = 2.143 mm).

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
