# Peer review of "140 GHz Ultra-Long Bessel–Like Beam with Near-Wavelength Beamwidth"

_sensors, 2020, doi:10.3390/s20236791_

Round 1
Reviewer 1 Report
In their manuscript, the authors report on the realization of finite-energy Bessel beams in the sub-terahertz range. Furthermore, they study how the propagation range can be extended by employing a defocus before their axicon lens. The presented work is well written and scientifically sound. However, the novelty in regards to Bessel beam shaping is relatively minor. This is why considerable additions have to be made before I can recommend a publication in MDPI Sensors. Down below, I will list my suggestions and points of criticism following roughly the order of the manuscript. Moreover, I am going state in angular brackets how important I regard the corresponding issue.
-
In the abstract, you provide the number “100mm” after mentioning the axicon lens. It is not clear what this number stands for. Either clarify or remove. [medium]
-
There is a typo in the first line of the introduction. When stating the THz-range, there is a dot instead of a dash. [medium]
-
At the beginning of section 2, you list several methods for the realization of long-distance Bessel beams. In this regard, the following publication might be of interest to you: https://doi.org/10.1002/lpor.201900103
-
In section 2, you provide several equations but you are hardly using them. What cone angle does your beam have after passing through the 110° axicon? What FWHM does that entail? And what is the theoretical minimum FWHM for your wavelength (i.e. for a cone angle of 90°)? Lastly, what theoretical beam length do you expect for a collimated situation? [important]
-
In Fig. 1, the labels for the subplots are in very odd places. It would probably be best to move them towards the top-left position, respectively. [medium]
-
I do not fully agree with the sentence in line 88. Yes, the apex angle of the axicon determines the cone angle of the beam and therefore the FWHM of the main lobe. And yes, to extend the beam length, you would typically employ the input beam width in order to keep the FWHM constant. However, your sentence reads as if the axicon apex angle (and therefore the beam cone angle) does not influence the length of the beam at all. This is obviously not the case. I’d suggest rephrasing this sentence to make sure the reader does not understand it wrong. [high]
-
In line 101, you mention that it was surprising to you that the beam became longer despite w0 became smaller from moving the collimating lens. Actually, if you do the math, this is not surprising at all. What your defocus does is it modifies the rays of the beam in such a way, that the paraxial ones are hardly touched at all, while the most outer rays become shallower towards the optical axis. And since the outer rays constitute the beam profile further down the propagation direction, this change in angle makes your beam longer. If you now consider geometrical optics the length of your beam will be roughly z = w / tan(alpha), with alpha being the local cone angle. With this at hand, you can analyze whether you expect the beam to become longer or not instead of leaving the reader with “it was surprising”. [very important]
-
Some general remarks on section 2 and possibly my major point of criticism: This section lacks a lot of information. What I have just explained in my previous bullet point is nowhere to be found in your manuscript. That is, how does extending the propagation range actually work. Please add this in some form or another. Personally, I would utilize several new plots in the discussion. First, how does the beam length change with the input beam waist? Second, how does the beam length change with the input beam defocus if w0 remains constant? It will become immediately clear, that this is the only way to create really long beams if you do not want to use gigantic optics. Third, what is to be expected in your setup when you move the collimating lens. This last plot would be mainly for the comparison with the experimental results later down the line. [very important]
-
Another important point that follows immediately from the previous one and which is nowhere to be found in your manuscript is the following: Because the outer rays of your beam (after introducing the defocus) propagate shallower towards the optical axis, the beam profile will scale larger the further the beam propagates. Hence, also the FWHM will increase. Again, this should be discussed (and experimentally verified) in one way or another. Personally, I would once more add a graph showing the FWHM along the propagation direction for several defocusing situations. You could then discuss how much beam expansion you are willing to bear. Maybe set the threshold at two times the wavelength or something like that. [very important]
-
The rendered visuals of Fig 2 look really nice. But I guess a technical drawing would be more educational to the reader [minor]
-
When you explain the experimental setup, you could add a short sentence mentioning why you cannot use a CCD camera. It may happen that people read your work who are unfamiliar with your wavelength range. [minor]
-
In section 4 (and possibly before) you are using the term “Bessel-Gauss” independently of whether you have introduced a defocusing phase or not. I’d suggest using a differentiating term like “prolonged” or “extended” so the reader is made aware that these beams are no longer strictly of Bessel-Gauss nature. [medium]
-
In line 141 you mention diverging side-lobes. What you are actually seeing is the scaling of the beam profile I mentioned before. Not only the side-lobes appear to propagate outwards. The main lobe also becomes larger. This part will probably need some modifications after section 2 has been reworked. [medium]
-
The sentences from line 142 onward were slightly confusing. I was not able to make out whether your beam was longer or shorter than expected. Maybe rephrase. [medium]
-
Figure 5 is way too busy. I’d suggest splitting it into 4 distinct panels. [medium]
-
When thinking about Figure 5, I realized that there is only one thing that makes your experiments really complicated and hard to predict. And this is that when you move your collimating lens you are not only changing beam angles but also the waist on the axicon. If you were to compensate for the latter, everything would become nice and straightforward. This could for instance be done by adding an aperture right in front of the axicon, limiting the beam diameter to the smallest one under consideration. Or you move the entire setup, which is probably much more difficult. [absolutely your decision; you would have to remeasure almost everything]
-
Figure 6 makes not a lot of sense considering what I have explained before. It needs to be replaced by a graph showing the FWHM along the propagation direction. And, of course, the result needs to be compared to the theoretical prediction. [important]
-
Table 2 will probably no longer be needed. But if you leave it in, please add theoretical values as well, so that they are easy to compare. [medium]
-
Checking the literature section, I’ve noticed almost one-third of quotations are self-citations. Please remove the non-relevant ones. [important]
-
Speaking of citations, please perform another literature search. I am pretty sure that someone else has already discussed the option to extend the length of Bessel beams by defocusing the input beam or modulating its phase in a more sophisticated way. I think the paper I mentioned before also partially dealt with that. But it was not its primary point. It would be negligent not reading and citing those other publications. The behavior of Bessel beams isn’t different for different regions of the electromagnetic spectrum. [important]
As stated in the beginning and certainly clear from my comments, this manuscript still needs a lot of work. If it would have been in the visible spectral range, I would have rejected it for lack of novelty. But I think the studied wavelength could make it interesting to some people. Therefore, I am willing to support a publication in MDPI Sensors after my points of criticism have been dealt with.
Author Response
We appreciate your elucidation of the deficient aspects of our manuscript in detail. We also think that the defocused beam-shaping method is not new. Indeed, it is not difficult to guess that the focusing beam length could be extended using the first lens shift. Rather than showing the beam-adjustment method itself, the focus of our paper is to investigate the dual characteristics of the focusing beam behind the axicon for practical purposes, including exposing how the length of the beam can be extended while maintaining a near-wavelength beamwidth. Therefore, we examined their properties using a custom-made experimental profiler to determine whether it was possible to implement both characteristics simultaneously and, if so, to what extent. Unfortunately, we did not clearly describe this motivation. Hence, As per your recommendation, we modified inappropriate expressions, notations, etc. Furthermore, additional calculations and their results were provided. Please see the attachment.
- In the abstract, you provide the number “100mm” after mentioning the axicon lens. It is not clear what this number stands for. Either clarify or remove. [medium]
è”100 mm” means the lens diameter, but “diameter” was omitted during the editing. Even if “100 mm” is removed, it is not a problem in context.
è100 mm => 100 mm
- There is a typo in the first line of the introduction. When stating the THz-range, there is a dot instead of a dash. [medium]
èThe typo was corrected.
è0.1.10 THz =>0.1–10 THz
- At the beginning of section 2, you list several methods for the realization of long-distance Bessel beams. In this regard, the following publication might be of interest to you: https://doi.org/10.1002/lpor.201900103
è Thank you for introducing valuable references. The “ring-lens” method and its references were added to the manuscript.
è narrow annular aperture filter, ring-lens, and computer-generated hologram
- In section 2, you provide several equations but you are hardly using them. What cone angle does your beam have after passing through the 110° axicon? What FWHM does that entail? And what is the theoretical minimum FWHM for your wavelength (i.e. for a cone angle of 90°)? Lastly, what theoretical beam length do you expect for a collimated situation? [important]
è The shortcomings found in Section 2 were accurately pointed out. The detailed discussion on the equations and theoretical values were added in Section 2, apart from one aspect:
The theoretical minimum beamwidth (1.02 mm ~ 0.475 wavelengths) at 140 GHz happens when total reflection occurs at the exit surface of the axicon (critical angle: arcsin (1/1.523)=41.04°), and it occurs when the half axicon angle equals 48.96°, where the DOF becomes zero. However, I think that this point is not matched with the other context, so that the part cannot be included in the manuscript.
In addition, some errors were also corrected in Section 2:
The refractive index of PE was 1.523. 1.54 was that of high-density polyethylene. Hence, it was corrected. In Eq. (3) of the manuscript, is the beam diameter. However, the equation is described for the beam radius. Thus, it should be corrected. The designed back focal length of the first lens was 72.25 mm exactly.
(3)
è Usually, with a thin axicon having a large apex angle, Eq. 1 is used to estimate the DOF. However, when the axicon apex angle becomes acute, a corrected equation should be used for accuracy:
. (4)
Here, is the overestimated length inside the axicon from the overall beam length ( ). is the beam radius on the incident surface of the axicon, whereas is on the exit surface. When the incident beam is collimated, and are the same, as shown in Fig. 1(a). In contrast, when the incident angle is not zero on the incident surface, the beam radius becomes different and varied with L variations. Moreover, is a critical parameter for calculating the precise DOF, as shown in Eq. (4) and Fig. 1(b).
|
(a) |
|
(b) |
Figure 1. Ray optics simulations (Opticstudio, Zemax, USA) for estimating the DOF of the Bessel–like beam: Bessel–Gauss beam generation (a) by a collimated beam on the axicon; (b) Bessel-like beam by a diverging beam on the axicon.
In the case where the incident beam on the axicon surface is collimated, using the above Eqs. (3) and (4) and the geometrical parameters, the DOF and beamwidth of the Bessel–Gauss beam can be easily estimated. For instance, can be approximated to 50 mm using Eq. (1), whereas is precisely calculated to 33 mm. It implies that is 17 mm, and is a more accurate value of the DOF. Angle is calculated as 25.9° by Eq. 2. In turn, the theoretical focusing beamwidth is calculated as 1.76 mm by Eq. 3.
Next, by further increasing the DOF from the collimated case, we employed a diverging beam on the axicon surface. Its incident angle on the axicon was manipulated by varying the distance (L) between the source flange and the first lens (designed back focal length = 72.5 mm, PE), where the axicon lens was spaced 60-mm away from the first lens surface. As a result of reducing the distance, L, from the collimated beam, Bessel-like beams having z-dependent cone angles can be generated, so that the DOF can be extended. Unfortunately, Eq. (2), which obtains the cone angle, , cannot be applied in this case. Hence ray-tracing simulations should be implemented for calculating the cone angle, , with varying distance L from the collimated beam, where the cone angle is that of the marginal ray. Using the calculated cone angles from the ray-tracing, the beam diameter, , and the exact DOF, , can be obtained from Eqs. (3) and (4). To calculate the , the beam radius, , to varied L should also be obtained from the ray-tracing. Additionally, to investigate the effect of the beam radius, , changes on the DOF variation, and the incident angle are shown in Fig. 1(a), also calculated from the simulations. In particular, is obtained from Eq. (4) based on the assumption that the incident beam radius, , is constant for varying L. By comparing and , the major factor to affect the extended DOF can then be examined.
When the incident beam is collimated, because the rays both adjacent to and away from the apex are parallel, the uniform focusing beam width of Eq. (3) is expected during beam propagation. In contrast, when increasing the incident angle on the axicon, the farther the rays are from the apex of the axicon, the corresponding cone angles become more z-dependent, as shown in Fig.1 (b). The cone angle, , decreases with the distance from the apex of the axicon, and Eq. (2) cannot be applied to the case, as discussed above. Moreover, Eq. (3) can be valid when the wavefront of the incident beam on the axicon is a plane wave [22]. On the contrary, this may be difficult to apply when the wavefront at the incident surface is spherical or parabolic. However, for roughly investigating the beamwidth change by the cone angles, it can be estimated with Eq. (3) by assuming that the wavefront on the incident surface is a plane wave.
For obtaining the full range of the z-dependent cone angles, the half-source divergence angle, q, was varied from 15° (marginal rays) to 1° (paraxial rays) at 2° increments for various lens distances, L. As a result, the range of marginal rays corresponding to each half-source angle can provide each cone angle. Thus, the FWHM variations on the propagation axis within the DOF range can be estimated for a given distance, L.
4.1. Ray-tracing simulation results on the Bessel-like beam shaping
Using the ray-tracing calculation of Section 2, both incident-beam radii on the incident and exit surfaces were calculated. Additionally, the incident angle on the axicon and the cone angle, , corresponding to the beam radius on the exit surface, was calculated for several distances, L, as shown in Figs. 3 (a) and (b). In particular, because the source was not a point, the distance, L, between the source flange and the first lens surface was defined as 67 mm in the case of the collimated beam, except that it was 5.5 mm from the designed back focal length of 72.5 mm. L ranged from 67 to 21 mm in 5-mm increments. Within this range, some L were identically adjusted to the measured distance. The calculation results are shown in Fig. 3.
Figure 3. Ray-tracing results for obtaining beam parameters: (a) beam-waist radius on the incident and exit surfaces of the axicon; (b) incident angle on the axicon surface and cone angle of the marginal ray; (c) DOF by varied , and with a constant ; (d) FWHM values for several internal cone angles, calculated with corresponding half-source angles (=q/2) , for varying distances, L, respectively
As expected, both the beam radius and the incident angle varied simultaneously on the axicon surface when the first lens distance, L, varied within the ranges. Fig. 3 (a) shows that, even if the lens distance, L, was decreased by one third from the collimated L=67 to L=21 mm beam, its radius, , on the exit surface would be reduced by only 2 mm (8%). Instead, the incident and cone angles of the marginal ray were rapidly changed, as shown in Fig. 3 (b). Using the beam radius and cone angle, the DOF values from Eq. (4) were calculated in Fig. 3 (c). As the distance, L, reduced from 67 to 21 mm, the cone angle decreased by 10.8°. Hence, the DOF of the Bessel-like beam doubled from 33 to 67 mm. Furthermore, when varying the L, the in Fig. 3 (c) was calculated based on the assumption that the beam radius, , on the exit surface was constant, whereas was calculated based upon a varied beam radius, . Fig. 3 (c) shows that two curves were almost identical, implying that the incident-angle change could be a major factor affecting the extended DOF as it corresponds to the reduced marginal cone angle.
Using Eq. (3) and the parameters of Figs. 3 (a) and (b), the focusing beamwidth was also estimated, as shown in Fig. 3 (d). When L=67 mm (collimated), the beamwidth was easily estimated using Eq. (3). However, when decreasing the lens distance, L, the cone angles within the range had a range of values (not singular), as discussed. To calculate the full ranges of the cone angle for a given distance, L, the half-source angles for each marginal ray were varied from 15 to 1° in 2° increments. Each marginal ray corresponding to the given half-source angle constructed the cone angles via ray-tracing. The FWHM was then obtained using Eq. (3). Among the results, the variations of FHWM for four internal angles is shown in Fig. 3 (d). At the collimated distance, L, all cone angles were parallel. Thus, all FWHM values were identical. In contrast, with a decreasing L, the non-parallel inner rays formed different cone angles, and the FWHM of each is shown in Fig. 3 (d). This shows that the FWHM difference between internal angles increased with a decreasing L. Thus, the farther away from the apex the axicon, the wider the beamwidth. To compare the experimental results, simulation results are discussed next. Although not shown in the graphs, in the extreme case where L was 0, the calculation result shows that the DOF was 94.8 mm (almost tripled), and the beam width as 4.08 mm (only 1.9 wavelengths). Therefore, the beamwidth theoretically does not exceed two wavelengths in tandem with more than twice the DOF.
.
- In Fig. 1, the labels for the subplots are in very odd places. It would probably be best to move them towards the top-left position, respectively. [medium]
è Per your suggestion, the position of the labels has been moved, and we are grateful that it looks much better. Refer to the text in response 4.
- I do not fully agree with the sentence in line 88. Yes, the apex angle of the axicon determines the cone angle of the beam and therefore the FWHM of the main lobe. And yes, to extend the beam length, you would typically employ the input beam width in order to keep the FWHM constant. However, your sentence reads as if the axicon apex angle (and therefore the beam cone angle) does not influence the length of the beam at all. This is obviously not the case. I’d suggest rephrasing this sentence to make sure the reader does not understand it wrong. [high]
è The sentence in line 88 seems to have a problem. We agree with the reviewer's opinion. Thus, the entire sentence was deleted and replaced with new content. Refer to the text in response 4.
- In line 101, you mention that it was surprising to you that the beam became longer despite w0 became smaller from moving the collimating lens. Actually, if you do the math, this is not surprising at all. What your defocus does is it modifies the rays of the beam in such a way, that the paraxial ones are hardly touched at all, while the most outer rays become shallower towards the optical axis. And since the outer rays constitute the beam profile further down the propagation direction, this change in angle makes your beam longer. If you now consider geometrical optics the length of your beam will be roughly z = w / tan(alpha), with alpha being the local cone angle. With this at hand, you can analyze whether you expect the beam to become longer or not instead of leaving the reader with “it was surprising”. [very important]
è We appreciate the kind explanation. Recognizing that it is an exaggerated expression to emphasize the effect of defocusing, the entire sentence was removed and replaced with new content suggested by the reviewer, including additional simulations. Refer to the text in response 4.
- Some general remarks on section 2 and possibly my major point of criticism: This section lacks a lot of information. What I have just explained in my previous bullet point is nowhere to be found in your manuscript. That is, how does extending the propagation range actually work. Please add this in some form or another. Personally, I would utilize several new plots in the discussion. First, how does the beam length change with the input beam waist? Second, how does the beam length change with the input beam defocus if w0 remains constant? It will become immediately clear, that this is the only way to create really long beams if you do not want to use gigantic optics. Third, what is to be expected in your setup when you move the collimating lens. This last plot would be mainly for the comparison with the experimental results later down the line. [very important]
èPer your remarks, we implemented additional calculations shown in response 4, which shows how the beam propagation works when varying the collimation lens distance. With four new plots (including beam length and beamwidth changes on beam waist and incident angle), a detailed discussion was provided in Sections 2 and 4 of the revised manuscript.
- Another important point that follows immediately from the previous one and which is nowhere to be found in your manuscript is the following: Because the outer rays of your beam (after introducing the defocus) propagate shallower towards the optical axis, the beam profile will scale larger the further the beam propagates. Hence, also the FWHM will increase. Again, this should be discussed (and experimentally verified) in one way or another. Personally, I would once more add a graph showing the FWHM along the propagation direction for several defocusing situations. You could then discuss how much beam expansion you are willing to bear. Maybe set the threshold at two times the wavelength or something like that. [very important]
è Thank you for helpful suggestion. FWHM along the propagation direction was simulated with ray-tracing for several defocusing distances. It is now described in Sections 2 and 4. Meanwhile, it might be necessary to set the threshold of the beamwidth. However, both DOF and beamwidth do not increase indefinitely, as described near the end of response 4. Thus, the discussion on the threshold value is not indicated in this study. Alternatively, calculations and results on the limited case were described in detail.
- The rendered visuals of Fig 2 look really nice. But I guess a technical drawing would be more educational to the reader [minor]
è The figures for understanding the basic concept have been revised in Fig. 1. Fig. 3 (formerly Fig. 2) is now a rendered drawing instead of a photograph of an actual measuring setup.
- When you explain the experimental setup, you could add a short sentence mentioning why you cannot use a CCD camera. It may happen that people read your work who are unfamiliar with your wavelength range. [minor]
è The following sentences were added to the manuscript.
èIn the visible and infrared ranges, the beam profiles could be easily measured using a commercially available beam-profiler (e.g., a charged coupled-device camera). However, because there was no method to measure the cross-sectional profile for a long-range (several tens of cm), a custom-made measurement setup was developed as follows.
- In section 4 (and possibly before) you are using the term “Bessel-Gauss” independently of whether you have introduced a defocusing phase or not. I’d suggest using a differentiating term like “prolonged” or “extended” so the reader is made aware that these beams are no longer strictly of Bessel-Gauss nature. [medium]
è We totally agree with the reviewer's opinion. In case of the collimated incident beam or the general description, we use the Bessel–Gauss beam as is, not the collimated incident beam. This was modified to the Bessel-like beam.
- In line 141 you mention diverging side-lobes. What you are actually seeing is the scaling of the beam profile I mentioned before. Not only the side-lobes appear to propagate outwards. The main lobe also becomes larger. This part will probably need some modifications after section 2 has been reworked. [medium]
èPer the reviewer's rigorous comments, it is correct that the main-lobe and side-lobe varies with DOF, and the side lobe itself does not diverge. This part has been modified to be more accurate.
è As expected in the ray-tracing simulation, the beamwidth of the Bessel-like beam broadened with the beam propagation, and the side-lobe peak positions varied. .
- The sentences from line 142 onward were slightly confusing. I was not able to make out whether your beam was longer or shorter than expected. Maybe rephrase. [medium]
è This has been modified to display the sentence more accurately without confusion.
è However, the measured DOF was more extended than in the simulation. .
- Figure 5 is way too busy. I’d suggest splitting it into 4 distinct panels. [medium]
èWe agree with the reviewer's suggestion and divided the figure into distinct panels. Reflecting the opinion of other reviewers, the six panels were displayed with the smoothing curves, including the collimated case.
Figure 7. Smoothed axial intensity profiles for estimating the DOF (L=67, 56, 51, 46, 41, and 36 mm). All curves were normalized to maximum intensity. Each DOF values of the figures are summarized in Table 2.
- When thinking about Figure 5, I realized that there is only one thing that makes your experiments really complicated and hard to predict. And this is that when you move your collimating lens you are not only changing beam angles but also the waist on the axicon. If you were to compensate for the latter, everything would become nice and straightforward. This could for instance be done by adding an aperture right in front of the axicon, limiting the beam diameter to the smallest one under consideration. Or you move the entire setup, which is probably much more difficult. [absolutely your decision; you would have to remeasure almost everything]
èAs discussed in manuscript, the beam radius at the exit surface reduced only by 2 mm, according to variation of the first lens distance from 67 to 21 mm. It implies that the DOF can be affected mainly by the incident angle on the axicon. Thus, even if the iris is not used in the setup, the beam-length change caused by the defocus can be sufficiently explained by referring to the above ray-tracing simulations. Thus, an additional experiment was not conducted.
- Figure 6 makes not a lot of sense considering what I have explained before. It needs to be replaced by a graph showing the FWHM along the propagation direction. And, of course, the result needs to be compared to the theoretical prediction. [important]
è Thank you for letting us know the additional aspects. The experimental FWHM values within the DOF range were expressed as several new graphs in Fig. 9. The numerical FWHM values were also described in Fig. 8 for comparison with the experimental ones.
Figure 8. FWHM variations along the Z-axis obtained from ray-tracing and Eq. (3) (L=67, 56, 51, 46, 41, and 36 mm). Reference line denotes the source wavelength (2.143 mm).
Figure 9. FWHM variations along the Z-axis directly obtained from measured 2D intensity profiles (L=67, 56, 51, 46, 41, and 36 mm). Reference line denotes the source wavelength (2.143 mm).
- Table 2 will probably no longer be needed. But if you leave it in, please add theoretical values as well, so that they are easy to compare. [medium]
è As suggested, theoretical values, such as DOF and beamwidth, were added to Table 2 to facilitate comparison.
Table 2. DOF and beamwidth obtained from various beam propagation profiles.
|
Lens distance (L) |
DOF |
Beamwidth (mean values) |
|
||||||
|
Ray tracing |
Measurement |
Ray tracing |
Measurement |
||||||
|
67 mm |
33 mm |
21 mm |
1.76 mm |
0.82 λ |
1.81 mm |
0.84 λ |
|
||
|
56 mm |
40 mm |
61 mm |
1.87 mm |
0.87 λ |
2.19 mm |
1.02 λ |
|
||
|
51 mm |
43 mm |
70 mm |
1.92 mm |
0.90 λ |
2.48 mm |
1.16 λ |
|
||
|
46 mm |
47 mm |
109 mm |
1.98 mm |
0.92 λ |
2.60 mm |
1.21 λ |
|
||
|
41 mm |
50 mm |
123 mm |
2.04 mm |
0.95 λ |
2.77 mm |
1.29 λ |
|
||
|
36 mm |
54 mm |
154 mm |
2.10 mm |
0.98 λ |
3.01 mm |
1.40 λ |
|
||
(λ: Source wavelength = 2.143 mm)
- Checking the literature section, I’ve noticed almost one-third of quotations are self-citations. Please remove the non-relevant ones. [important]
èWe agree with the reviewer, and the less relevant references were deleted.
- Speaking of citations, please perform another literature search. I am pretty sure that someone else has already discussed the option to extend the length of Bessel beams by defocusing the input beam or modulating its phase in a more sophisticated way. I think the paper I mentioned before also partially dealt with that. But it was not its primary point. It would be negligent not reading and citing those other publications. The behavior of Bessel beams isn’t different for different regions of the electromagnetic spectrum. [important]
è Additional references on the DOF extension of the Bessel beam were added to the reference section.
- Vetter, C.; Steinkopf, R.; Bergner, K.; Ornigotti, M.; Nolte, S.; Gross, H.; Szameit, A., Realization of Free‐Space Long‐Distance Self‐Healing Bessel Beams. Laser Photonics Rev. 2019, 13, 10, 1900103.
- Belyi, V.; Forbes, A.; Kazak, N.; Khilo, N.; Ropot, P., Bessel–like beams with z–dependent cone angles. Opt. Express 2010, 18, 3, 1966–1973.
- Saikaley, A.; Chebbi, B.; Golub, I., Imaging properties of three refractive axicons. Appl. Opt. 2013, 52, 28, 6910–6918.
- Yao, Z.; Jiang, L.; Li, X.; Wang, A.; Wang, Z.; Li, M.; Lu, Y., Non-diffraction-length, tunable, Bessel-like beams generation by spatially shaping a femtosecond laser beam for high-aspect-ratio micro-hole drilling. Opt. Express 2018, 26, 17, 21960–21968.
- Brzobohatý, O.; Čižmár, T.; Zemánek, P., High quality quasi-Bessel beam generated by round-tip axicon. Opt. Express 2008, 16, 17, 12688–12700.

Reviewer 2 Report
The draft has described a CW sub-terahertz Bessel-Gauss beam generation and characterization using an axicon lens. By setting a proper apex angle of the axicon, the authors measured the depth of focus (DOF) and the focusing beamwidth at different beam divergence. The prospect of the beam is for the food inspection and computer tomography. The draft is written well and the research is presented clearly. But I don’t feel the studies are important or novel enough to be presented at sensors in the current version. Here are my comments.
The authors have demonstrated a long DOF of a Bessel-Gauss beam by changing the beam divergence. In principle, it follows the equation from Ref. [6], which is published by the authors themselves several years ago. And the general setup is also shown in Ref. [6]. The beam divergence can bring a different DOF is not a surprise to readers even though it is done using a sub-terahertz Bessel-Gauss beam. Changing the beam divergence by adjusting the distance between lens is like a practice from the setup in Ref. [6]. Considering the application prospect, according to the intensity profile, it looks like pretty noise along z direction. Even though the authors showed the normalized intensity at several position, I am still wondering how the beam profile looks like along z, which is very important if it is used in practice. Moreover, I also have several suggestions for presentation I think Fig.1 does not need two sub figures. People will understand how the difference between collimated beam and diverging beam. Fig.5 is very messy, and it is difficult to get information. In conclusion, I do not see enough studies to show the application of the long DOF beam in the current version. I would suggest authors to include more evidence in practice.
Reviewer 3 Report
The paper presents a method to extend the depth of focus of an axicon formed Bessel-Gauss-GHz-Beam. The varied parameter is the position of a lens in front of the axicon. With the extended focus, it shell become possible to generate high quality images in non-inversive material tests.
In the introduction the application of using THz and GHz waves for non-inversive food quality inspection is described. The style of writing is very well and the line of arguments mentions several important points, but the most important statements (e.g. the characteristics of THz beam shaping) are just mentioned at the end and they are not put into focus. Instead there is a very long paragraph about the applications in the food industry, which has not much to do with what was investigated in this paper. All in all, the first part of the introduction should be shortened a bit and the basics of beam shaping should be described more detailed.
In the simulation chapter several important information are missing. For the source, no parameter but the wavelength is given. But when investigating the influence of the lens behind the source, at least the diverging angle must be given. Otherwise, the whole paper has no scientific value. Also,
other important geometric parameters should be added to figure 1 (distance between the lenses/axicons, …). In this figure, the interesting part of the beam is shown much too small. Instead there is a second axicon, a second lens and a detector, which is not mentioned in this chapter and for the simulation irrelevant. The sentence “[…], the apex angle should be carefully chosen because it is proportional to the beam spot size, […]” should be corrected. It cannot be true, that there is proportional correlation between an angle and a length.
In figure 2, many parameters of the used components are given. In the end, it is not clear, what is the resolution of the whole measuring setup. Is it limited by the pinhole of the detector or by the resolution of the stage? Again, in this chapter, important details in the figure like the detector are much too small.
In chapter 4 the results of the measurement are discussed. Figure 3 gives a good overview on the scanning strategy. But for showing the diverging of the higher orders, the used colours should be changed. The mentioned oscillatory interference patterns should be handled with care. It seems like the reason for them is not clearly understood. So, it is not proven, that the smoothed curve shows the truth. It is not explained, why there are some regions with higher interference effects and some with lower ones. However, the smoothed curves do not look like a typical Bessel-Gauss-beam-profile. In figure 5, it shows multiple maxima for some of the curves. This is untypical for axicon applications, but not discussed in the text. When coming to the radial distribution, the values in table 2 do not fit to the diagram in figure 6. In the diagram the “L=46 mm”-curve has the smallest beam width; in the table it is the “L=51 mm”-experiment. Further, it is not explained, why just four values for L has been tested. Concluding chapter 4, too many points are ignored in the discussion, so the significance of these results is questionable.
All in all, it has been proven, that there is a way to elongate the depth of focus for GHz Gauss-Bessel-beams. Whether this can help for the application in the food industry, is not yet clarified. Considering, that the whole results are based on moving a lens back and forth, the scientific value of the paper is not very high until there are more detailed investigations about the propagation of the beam behind the axicon. For publishing this paper, several major revisions should be done.
See also my comments in the attached manucript.

Round 2
Reviewer 2 Report
I appreciate the effort of the authors. The revised manuscript has provide more information and studies both the experimentally and theoretically. I would like to recommend the publications.
Reviewer 3 Report
All my criticisms and comments were satisfactorily implemented in the revised version. The manuscript can be accepted for publication in its present form.